# Deep learning-based age estimation from clinical Computed Tomography image data of the thorax and abdomen in the adult population

**Bjarne Kerber** [1]*, **Tobias Hepp**[2], **Thomas Küstner**[1], **Sergios Gatidis**[1,2]

**1** Department of Diagnostic and Interventional Radiology, University Hospital Tuebingen, Tuebingen, Germany, **2** Max Planck Institute for Intelligent Systems, Tuebingen, Germany

* bjarne-jonas.kerber@student.uni-tuebingen.de

## Abstract

Aging is an important risk factor for disease, leading to morphological change that can be assessed on Computed Tomography (CT) scans. We propose a deep learning model for automated age estimation based on CT- scans of the thorax and abdomen generated in a clinical routine setting. These predictions could serve as imaging biomarkers to estimate a "biological" age, that better reflects a patient's true physical condition. A pre-trained ResNet-18 model was modified to predict chronological age as well as to quantify its aleatoric uncertainty. The model was trained using 1653 non-pathological CT-scans of the thorax and abdomen of subjects aged between 20 and 85 years in a 5-fold cross-validation scheme. Generalization performance as well as robustness and reliability was assessed on a publicly available test dataset consisting of thorax-abdomen CT-scans of 421 subjects. Score-CAM saliency maps were generated for interpretation of model outputs. We achieved a mean absolute error of 5.76 ± 5.17 years with a mean uncertainty of 5.01 ± 1.44 years after 5-fold cross-validation. A mean absolute error of 6.50 ± 5.17 years with a mean uncertainty of 6.39 ± 1.46 years was obtained on the test dataset. CT-based age estimation accuracy was largely uniform across all age groups and between male and female subjects. The generated saliency maps highlighted especially the lumbar spine and abdominal aorta. This study demonstrates, that accurate and generalizable deep learning-based automated age estimation is feasible using clinical CT image data. The trained model proved to be robust and reliable. Methods of uncertainty estimation and saliency analysis improved the interpretability.

## 1. Introduction

The chronological age of a patient is an important risk factor for cardiovascular, oncological and neurodegenerative disease [1–3], influencing almost every medical decision from clinical examination to further diagnostics, therapy and aftercare.

However, the aging process is characterized by an extreme interindividual heterogeneity because of varying genetic constitution, way of life and environmental factors [4], making the

**Data Availability Statement:** The test dataset used for technical validation can be accessed publicly from https://wiki.cancerimagingarchive.net/pages/viewpage.action?pageId=93258287. However, the

in-house dataset used for model training comprises sensitive medical image data, and as such is subject to strict patient privacy and confidentiality regulations. Unfortunately, the authors are unable to publicly share the data due to these legal and ethical restrictions. The authors encourage researchers who are interested in replicating or validating the study's findings to contact the corresponding author for further information and potential collaborations. Access to the in-house dataset can be granted to qualified researchers who meet the necessary criteria and comply with the applicable data protection and privacy regulations. Data access requests can also be sent to: University Hospital of Tuebingen, Diagnostic and Interventional Radiology, Medical Image and Data Analysis (MIDAS.lab), Otfried-Müller-Str. 3, 72016 Tübingen.

**Funding:** Deutsche Forschungsgemeinschaft (DFG, German Research Foundation) under Germany's Excellence Strategy – EXC 2180 – #390900677 and EXC 2064/1 - #390727645.

**Competing interests:** The authors have declared that no competing interests exist.

chronological age a rather unreliable parameter to estimate a patients true biological condition. Thus, it has been tried in numerous studies to identify biomarkers to estimate a "biological" age.

Age predictions from deep learning models trained on medical imaging data of a healthy cohort correlate to the hypothetical biological age and can be used as an imaging biomarker [5]. Differences of the predicted age and chronological age in brain magnetic resonance imaging (MRI) scans are suspected to imply pathologies like neurodegenerative disease [6,7], schizophrenia [8] or diabetes [9]. A higher predicted age based on Chest Radiographs also correlated significantly with a higher long-term all-cause and cardiovascular mortality [10].

Accurate automated biological age estimation could help physicians in better assessing the true physical condition of a patient. For instance, the estimated biological age could replace the chronological age as a parameter in age-dependent risk scores, possibly improving their accuracy. Furthermore, the concept of biological age itself might be more understandable for patients than abstract scores [11].

Yet, the use of deep learning models in the critical infrastructure of healthcare requires comprehensible and reliable predictions. Deep learning models today are often black-box-like systems [12]. Thus, ambiguous or out-of-distributional inputs might lead to predictions that are not trustworthy [13] without the user of the model knowing or understanding the causes of failure.

One approach to engage these challenges is the implementation of a deep learning model, that is able to not only deliver an accurate prediction but also quantify its uncertainty over it [13].

Uncertainty arises from two sources: A lack of knowledge, called "epistemic" (modelling) uncertainty, and the "aleatoric" (stochastic) uncertainty, that is influenced by the inherent random noise [14]. In the case of age estimation, aleatoric uncertainty arises from the physiological variability of subjects and the aging process. Visually similar image data might be present at different chronological ages [15].

To this day, uncertainty for automated age estimation has only been explored in few studies. Shi, Yan [16] implemented epistemic and aleatoric uncertainty estimation for fetal brain age estimation using MRI data. Becker, Klein [17] proposed to use the estimated uncertainty instead of the difference between age estimation and predicted brain age to differentiate between healthy and diseased subjects.

Another approach to improve the interpretability of predictions and gain insights into the black box is the creation of visual explanations. Several methods of Class Activation Mapping (CAM) can be used to identify the most influential subregions in the input imaging data that lead to the prediction. Langner, Wikström [18] identified the aortic arch and knee as indicators of aging using Grad-CAM [19] and a deep learning approach trained on a whole-body MRI dataset. Raghu, Weiss [10] showed the importance of the mediastinum for age estimation using chest radiographs.

Although CT image data, created in the process of clinical routine, is widely available, most studies in the past preferred MR imaging for automated age estimation outside of bone age assessment [5,20]. While many previous studies used highly standardized image data from large cohort studies, clinical imaging data is much more heterogenous.

The contributions within this work are as follows. We propose a deep learning model for automated age estimation based on CT-scans of the thorax and abdomen generated in a clinical routine setting. The aleatoric uncertainty was modeled using a heteroscedastic approach. Generalization performance as well as reliability and robustness of our proposed model were assessed using a publicly available test dataset. Finally, body regions that are important for age estimation were identified using saliency maps generated with Score-CAM.

## 2. Methods

### 2.1 Study population

In this study, we used image data derived from two sources: An in-house dataset for training and validation and a publicly available dataset [21] for testing of model performance and generalization.

The ethics committee of the University Hospital Tübingen gave their consent for the retrospective analysis of the personal and image data of the patients included in the in-house dataset (Project number 387/2020BO, vote of 03.06.2020). Personal information was desensitized, patient names were pseudonymized in this work. Owing to the retrospective nature, the requirement for informed consent was waived in this study.

2419 soft tissue thorax and abdomen images of patients who, between 2010 and 2019, received an axial thorax/abdomen CT-scan in the emergency room, in which no pathological findings were reported, were drawn from the image archives of University Hospital Tübingen. 354 CT images of subjects younger than 20 or older than 85 were excluded, because of a lack of subjects outside of this age range in our dataset, as well as 412 images that included obvious pathologies or artifacts that made the image data unsuitable for age estimation, or were from a patient already included in the dataset. Only a single study of each subject was included. Of the 1653 patients included, 1120 (67.8%) were male, whereas 533 (32.2%) were female. The chronological age distribution of the study population and a flowchart of the inclusion and exclusion criteria for the training and validation and test set are shown in Fig 1.

Mean age of the study population in the training and validation dataset was 53.75 ± 19.46 years. Male subjects had a mean age of 52.04 ± 19.2 years, while female subjects had a mean age of 57.3 ± 19.57 years.

The test dataset contained CT-scans drawn from a publicly available Positron Emission Tomography (PET)/CT-dataset consisting of patients with PET-positive malignant lesions and of negative controls [21].

From the 900 subjects, a single examination of 422 subjects with a scan negative for malignancy was included in the test dataset. Subjects with a scan positive for malignancy were excluded. 199 subjects were female, whereas 222 were male. Female subjects were aged 59.23 ± 14.52 years on average, male subjects 59.58 ± 14.47 years.

For subjects included in the test dataset with more than one examination available, a second study was chosen to examine the reliability of network predictions (see 2.6), but was not used to evaluate generalization performance.

### 2.2 Image data

The 1653 CT image studies of the training and validation dataset were acquired using varying CT-Scanners and Reconstruction Kernels with a standardized slice thickness of 3 mm in the time period between 2010 and 2019.

The 421 CT image studies of the test dataset were acquired on a single CT-scanner (Siemens Biograph mCT) with a slice thickness of 2–3 mm between 2014 and 2018 [21].

Imaging characteristics are shown in Table 1.

### 2.3 Preprocessing

Image data was first converted from DICOM- to Nifti-Format and then resampled using cubic interpolation to a common shape of 112x112x224 voxel. To remove artifacts like stretchers or the CT examination table, the patient's body was segmented from the image utilizing Sobel

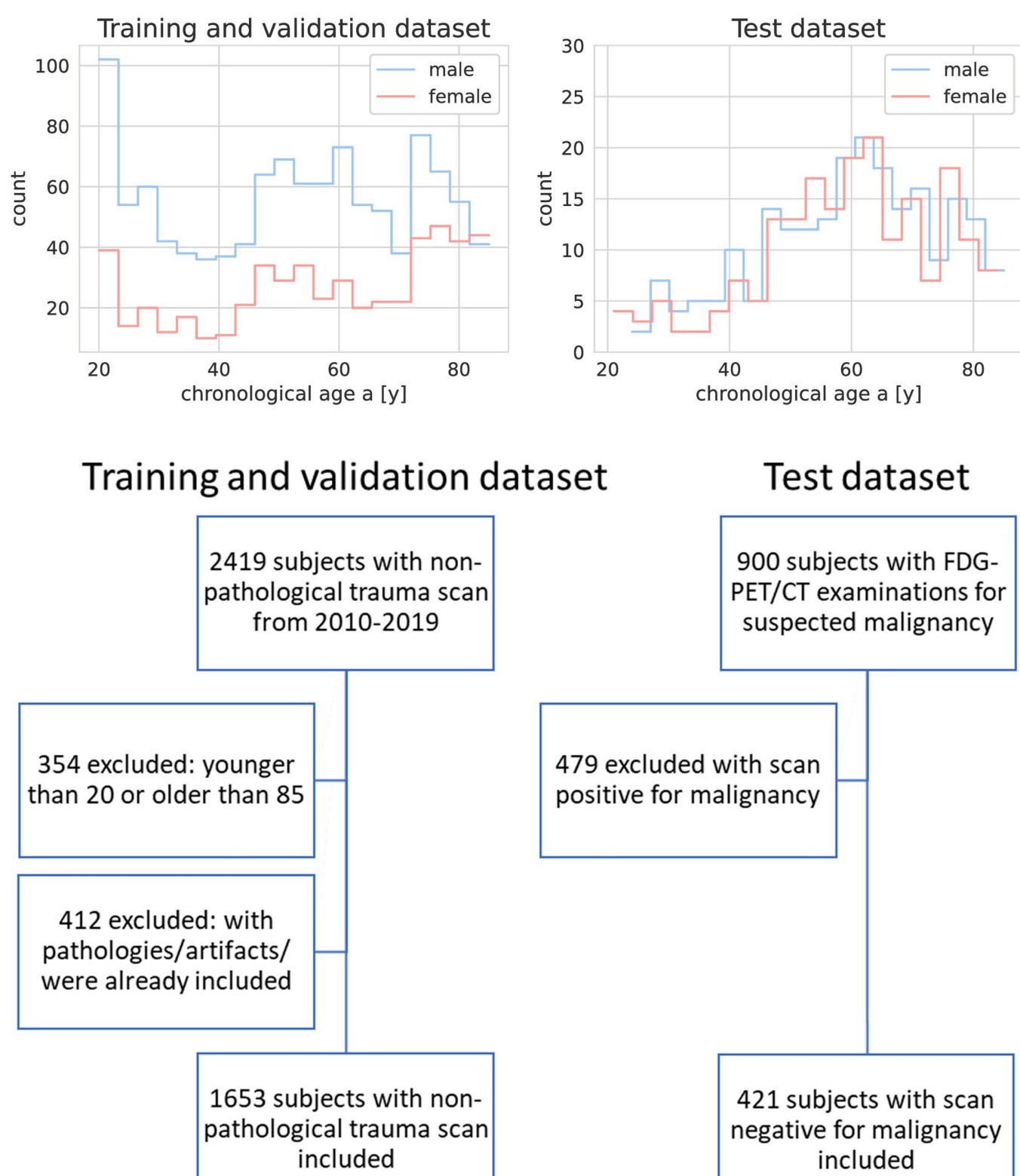

**Fig 1. Chronological age distribution of the study populations and inclusion and exclusion criteria of the training and validation dataset and the test dataset.**

**Table 1. Imaging characteristics of the training and validation (emergency trauma scans) and the test dataset (oncological staging) relevant to biases.** Slice thickness in emergency trauma scans was standardized at 3mm, while it ranged between 2–3 mm in the oncological staging scans. The share of female subjects was smaller in the training and validation dataset and they were on average older thank the male subjects. Subjects in the test dataset were on average older than their counterparts in the training and validation dataset.

| Indication for CT | Patient sex | Device | Kernel | n/o studies | age [y] |
|---|---|---|---|---|---|
| Training and validation dataset: emergency trauma scan w/o pathological findings | Female | SIEMENS SOMATOM Definition AS+ | I31f | 139 | 55.81 ± 19.21 |
| | | SIEMENS SOMATOM Force | Bf40d | 21 | 64.05 ± 13.73 |
| | | | Br40d | 19 | 56.68 ± 19.37 |
| | | SIEMENS Sensation 64 | B31f | 198 | 56.95 ± 19.86 |
| | | | B40f | 113 | 58.63 ± 20.98 |
| | | | B41f | 21 | 56.52 ± 18.39 |
| | | Other | Other | 22 | 57.23 ± 17.42 |
| | | **in total** | | **533** | **57.27 ± 19.53** |
| | Male | SIEMENS SOMATOM Definition AS+ | I31f | 311 | 52.92 ± 18.82 |
| | | SIEMENS SOMATOM Definition Flash | I31f | 10 | 56.90 ± 11.45 |
| | | SIEMENS SOMATOM Force | Bf40d | 39 | 56.49 ± 17.27 |
| | | | Br40d | 39 | 59.67 ± 18.52 |
| | | SIEMENS Sensation 64 | B31f | 344 | 50.42 ± 20.17 |
| | | | B40f | 313 | 52.04 ± 18.90 |
| | | | B41f | 39 | 50.00 ± 20.60 |
| | | Other | Other | 25 | 47.36 ± 14.31 |
| | | **in total** | | **1120** | **52.07 ± 19.20** |
| | **in total** | | | **1653** | **53.75 ± 19.46** |
| Test dataset: oncologic staging w/o evidence of active disease | Female | SIEMENS Biograph mCT | I31f | 199 | 59,23 ± 14.56 |
| | Male | SIEMENS Biograph mCT | I31f | 222 | 59,58 ± 14.51 |
| | **in total** | | | **421** | **59.32 ± 14.62** |

edge detection followed by watershed segmentation with an empirically chosen threshold of -600 Hounsfield units.

Then, frontal and sagittal thin-slab maximum intensity projections (MIP) of the central 20 voxel-layers were computed and, after Data Augmentation, concatenated into a 2D-Image with a common shape of 224x224 pixel.

Offline data augmentation was performed on the training dataset by randomly rotating the image between -5 and 5˚, cropping random patches sized 56x112 voxel and randomly resizing them in a size range between 1 and 1.5. Residual space was filled with zero-values. For each original image, 49 augmented images were created. Examples of augmented images are shown in Fig 2.

## 2.4 Network implementation and heteroscedastic noise model

A ResNet18-architecture [22] pretrained on the ImageNet-dataset by Pytorch [23], was modified to return two output channels predicting the mean $\mu(x)$ and standard deviation $\sigma(x)$, respectively $log(\sigma(x))$ of a Gaussian distribution. The network estimates a subjects age based on the input CT image $x$ (Fig 2).

Heteroscedastic aleatoric uncertainty was modeled using the Gaussian negative log likelihood loss [24] as empirical risk minimization objective.

For a batch $B$ consisting of n samples $B = \{(xi, ai), i = 1, \ldots, n\}$ where $x_i$ is the input CT-image and $a_i$ the corresponding chronological age, a convolutional neural network (CNN) with parameters $\theta \in \Theta$ predicts with its two output channels for each element the mean $\mu^\theta(x_i)$ and standard deviation $\sigma^\theta(x_i)$ of a gaussian distribution. The corresponding gaussian negative

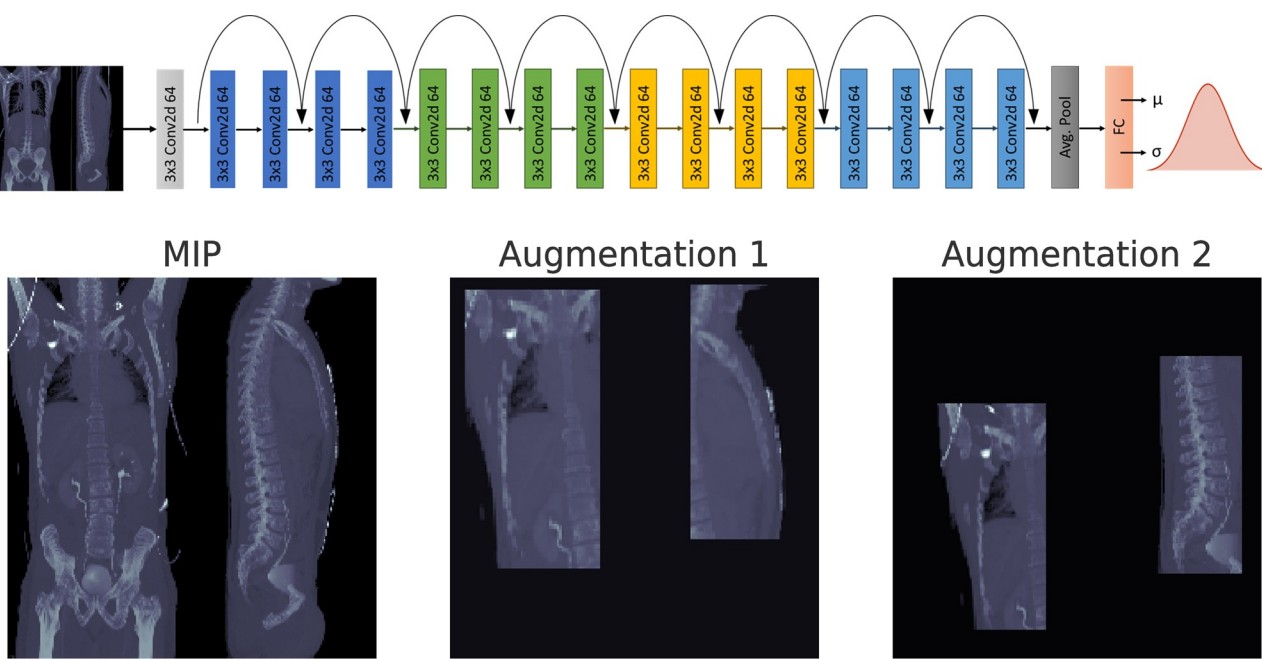

**Fig 2. Architecture of proposed model and exemplary input data.** Top: Resnet18 model architecture for age estimation. Bottom: Example original input image and augmented versions. Predicted age distributions are parameterized by the two output channels μ(x) and σ(x), respectively log(σ(x)) for a given input image x.

log likelihood loss is then defined as [25]:

$$L(B, \theta) = \frac{1}{n} \sum_{i=1}^{n} \log(\sigma^{\theta}(x_i)) + \frac{(a_i - \mu^{\theta}(x_i))^2}{2\sigma^{\theta}(x_i)^2} \qquad (1)$$

The model was implemented using the Pytorch Framework [23]. Models were trained and evaluated using 5-fold patient-leave-out cross-validation. The training and validation dataset was split 80%/20% into training and validation datasets along subjects for each fold, thus using 1322 original, respectively 66100 augmented images for training and 331 for validation. Hyperparameter optimization was performed by trial and error on the first cross-validation fold. An AdamW optimizer was used. The learning rate was set to $10^{-4}$ with a batch size of 32. Training data was augmented offline as stated above (2.3 Preprocessing). After 75 training epochs, the models with the smallest MAE during validation were chosen for each fold. Then, testing was performed on the independent test dataset described above (2.2). For the test predictions, the single predictions of the 5 cross-validation models were averaged. Training, validation and testing were performed on a Nvidia V100 graphics processing unit (GPU).

Public code is available on www.github.com/BjarneKerber/age_estimation/

## 2.5 Visual explanations

To identify subregions, that were most influential for the prediction of the network, Score-CAM [26] saliency maps were generated. In contrast to other procedures, like Grad-CAM, Score-CAM does not rely on the gradient information flowing into the last convolutional layer. Instead, Score-CAM uses the activation maps obtained from the last convolutional layer after the first forward pass to mask the input image. For each of the obtained activation maps *A'* masking the original image, another forward pass through the image is performed

generating a new activation map $A"$ each time. The result is then generated by a linear combination of score-based weights and the activation maps $A"$ using the rectified linear unit activation function [26]. Score-CAM showed superior results compared to GradCAM in previous studies [27]. The implementation by Früh, Fischer [27] was chosen. The generated saliency maps were assessed qualitatively.

## 2.6 Technical validation

Technical validation was performed to ensure robustness, reproducibility and generalizability of our proposed model, as suggested by Recht, Dewey [28]. Generalizability was assessed by testing our dataset on the independent test dataset described above (2.2). Robustness was verified by comparing model predictions on original images with model predictions on perturbated images of said test dataset. Modification of the normalized images (value range between 0 and 1) was done by adding random intensity variations (±0.1, drawn from uniform distribution), resizing of the original image data (zoom factor between 0.8 and 1.2, drawn from a uniform distribution) and injection of random gaussian noise (drawn from a gaussian distribution with mean 0 and sigma 0.01). Reliability was assessed by comparing model predictions on different imaging studies of the 139 subjects from the test dataset, for which more than one examination was available.

## 2.7 Statistical analysis

Results are given with mean values ± standard deviations. Model predictions and estimated uncertainty were analyzed with regard to used reconstruction kernels, CT-scanners and exposure as well as for subject sex and age. For group comparisons, the two-sample t-test and Kruskal-Wallace test were used. For correlation analysis, Pearsons r and $R^2$ values were used. Global significance level was set at 0.05.

## 3. Results

### 3.1 Age estimation

The achieved MAE after 5-fold cross validation was 5.76 ± 5.17 years averaged over the validation predictions of the trained networks, with a corresponding $R^2$ of 0.84 between true and predicted age. The subjects chronological age and corresponding predicted age are shown in Fig 3.

No significant difference in the absolute prediction error between female (MAE 6.07 ± 5.49 years) and male (MAE 5.60 ± 5.01 years) subjects was observed. The prediction error of female subjects (mean prediction error -1.36 ± 8.08 years) was significantly (p = 0.02) more negative than the prediction error of male subjects (mean prediction error -0.10 ± 7.50 years). No significant difference in the absolute prediction error was observed between different reconstruction kernels and CT-scanners. Absolute prediction errors of older (age > = 53 years) and younger subjects (age < 53 years) showed no significant difference. A significant (p<0.01), although very weak negative correlation (r = -0.085) between exposure (mAs) and absolute prediction error was observed.

The achieved MAE on the test dataset was 6.50 ± 5.18 years with a corresponding $R^2$ of 0.74. No significant difference in MAE (6.96 ± 5.45 vs. 6.07 ± 4.91 years) or prediction error between female and male subjects (-3.57 ± 8.09 vs. -2.56 ± 7.39 years) was observed.

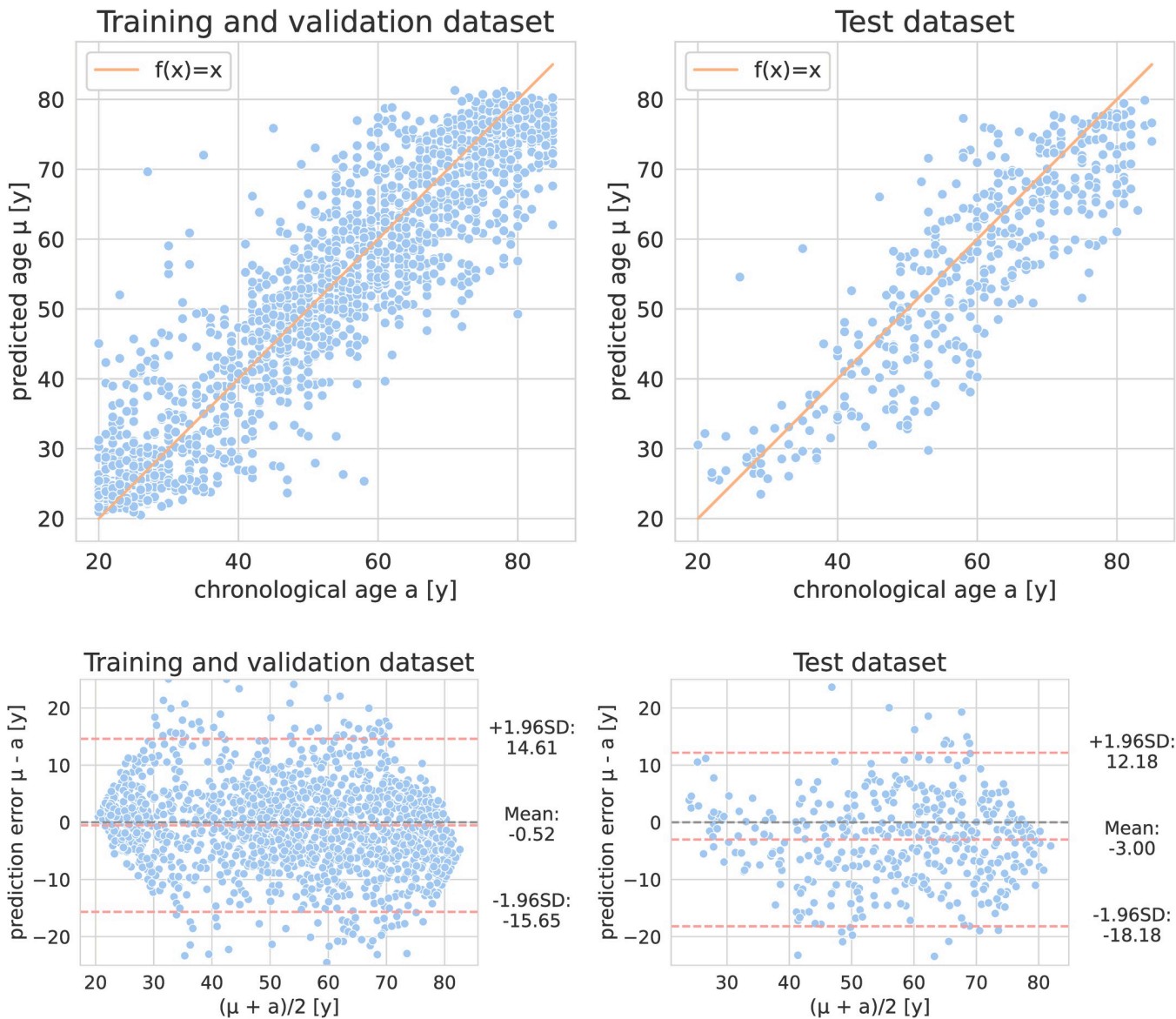

**Fig 3. Relation of model predictions with true chronological age.** Top: Scatter plot for predicted age μ versus chronological age a on all subjects of the training and validation and the test dataset. Bottom: Bland-Altman plots for the predicted age μ versus chronological age a of the training and validation and the test dataset. The orange line characterizes the function f(x) = x. Slight overestimation was overserved in younger subjects, and slight underestimation in older subjects of the training and validation dataset. The age of, especially older, subjects in the test dataset seemed to be slightly underestimated.

## 3.2 Uncertainty estimation

The mean uncertainty σ in the validation dataset was estimated as 5.01 ± 1.44 years. There was no significant difference in estimated uncertainty observed between female (5.12 ± 1.41 years) and male (4.96 ± 1.45 years) subjects, while older (> = 53 years) subjects had a significantly (p<0.01) lower estimated σ (4.86 ± 1.32 years) than younger (<53 years) subjects (5.20 ± 1.55 years).

We observed a significant moderate correlation (r = -0.40, p<0.01) between the number of examples per age and the mean uncertainty σ, while there was also a significant, though weak correlation between MAE and predicted uncertainty observed (r = 0.25, p<0.01). The absolute prediction error |μ-age| by estimated uncertainty σ is shown in Fig 4.

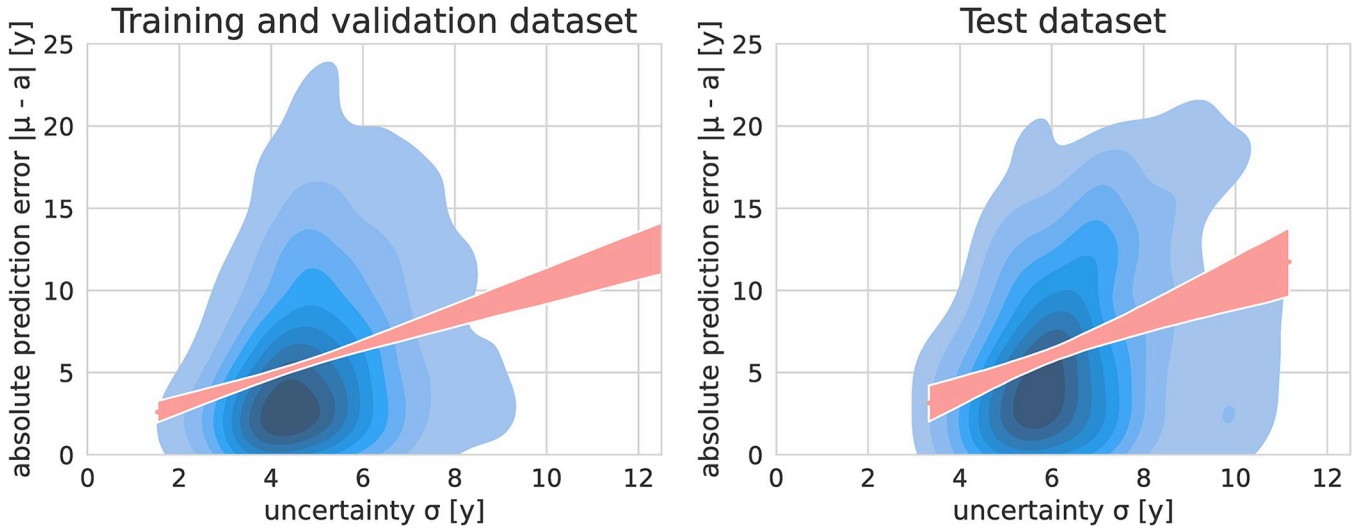

**Fig 4. Kernel density estimate plot for absolute prediction error versus predicted uncertainty of the cross-validation and test dataset.** A significant correlation between absolute prediction error and predicted uncertainty was observed for both datasets. The calculated regression line is shown in orange.

Regarding acquisition parameters, there was only a significant ($p<0.01$), although very weak correlation ($r = -0.045$) between exposure (mAs) and estimated uncertainty observed.

On the test dataset, mean uncertainty was calculated as $6.39 \pm 1.46$ years. Male subjects showed a mean uncertainty of $6.16 \pm 1.32$ years, with $6.64 \pm 1.57$ years for female subjects. The uncertainty was significantly higher for female subjects ($p<0.01$). A significant though weak correlation was observed between the MAE and uncertainty on the test dataset ($r = 0.31$, $p<0.01$).

The estimated uncertainty was significantly higher for subjects in the test dataset than in the cross-validation predictions made with the training and validation dataset ($p<0.01$).

The predicted uncertainty is plotted against the chronological age of the subjects in Fig 5.

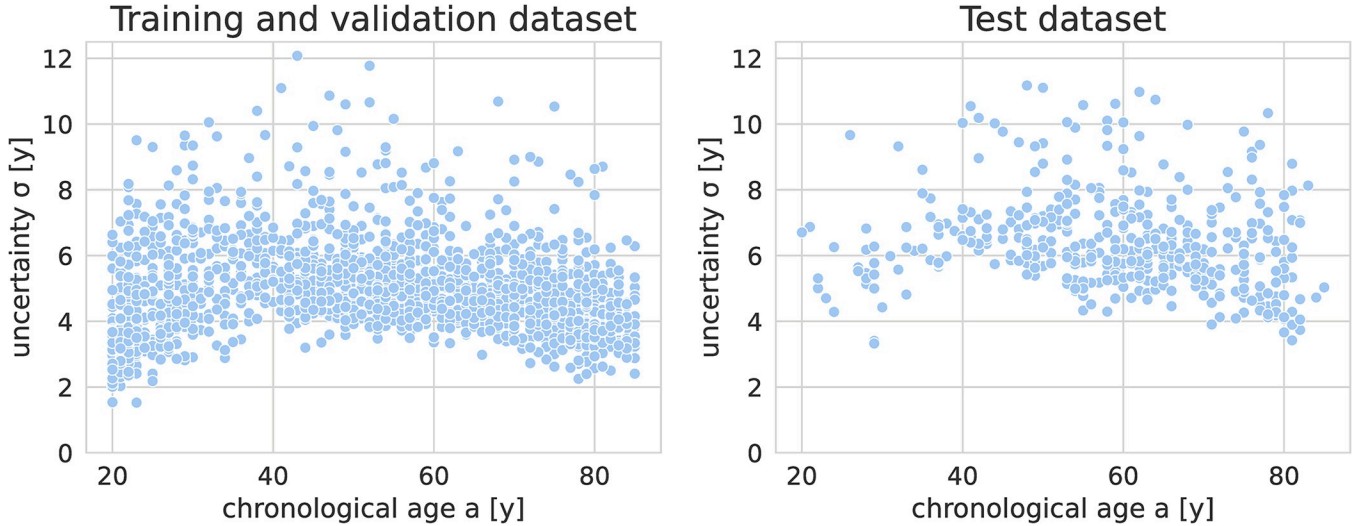

**Fig 5. Scatter plot for predicted uncertainty versus chronological age of the cross-validation and test dataset.** Younger and older patients show slightly lower uncertainty. The uncertainty seems to coincide with the number of subjects of a given age (see Fig 1). The uncertainty wase higher on the test dataset.

### 3.3 Visual explanations

Exemplary generated saliency maps are shown in Fig 6. The observed patterns, which were assessed qualitatively, seemed to be more consistent on the sagittal projection than in the frontal projection, where highlighted regions varied more. Over all age groups and models, the lumbar spine and abdominal aorta were highlighted as important for age estimation. This pattern occurred even in subjects where for example the legs were included in the image and thus differed significantly from the majority of subjects. To a lesser extent, abdominal organs like the kidneys were highlighted in the frontal projection, whereas thoracal structures seemed not to be part of the activated regions in most cases. The generated saliency maps for the test dataset showed a pattern similar to the saliency maps of the validation datasets.

### 3.4 Technical validation

The results for generalizability assessment on the independent test dataset are shown in section 3.1 and 3.2, especially in the Figs 3 and 5.

When reliability was assessed, model predictions for different studies of the same subject in the test dataset correlated strongly and significantly with an $R^2$ of 0.89.

When robustness was assessed, a strong, significant correlation between the age predictions for the original and modified images of subjects from the test dataset ($R^2 = 0.88$) was observed. There was no significant difference in MAE. The uncertainty for the modified images was significantly higher than for original images (7.02 ± 1.78 y vs. 6.39 ± 1.46 p<0.001).

The results of the reliability and robustness checks are shown in Fig 7.

## 4. Discussion

In this work, a deep learning-based model was trained for automated age estimation using CT image data of the thorax and abdomen acquired in a clinical setting. Uncertainty predictions and visual explanations in form of saliency maps were calculated. Testing was performed on a publicly available independent dataset.

It was shown, that accurate automated ages estimation is feasible, with a mean average error of 5.75 years and a $R^2$ of 0.84 on the training and validation dataset of 1653 subjects aged between 20 and 85 years. The aforementioned study carried out on whole-body MRI data of 32000 subjects aged between 44 and 82 from the UK Biobank study [18] reached a MAE of 2.49 years with a $R^2$ of 0.83 on a much smaller age range. A ResNet50 trained on 224316 chest radiographs from the CheXpert dataset [29] reached a MAE of 4.96 years, but with a $R^2$ of 0.94 [30]. In a study conducted by Azarfar, Ko [31] in subjects from a lung screening study with a smaller age range between 55 and 75 years, an MAE of 1.84 years was achieved with an inception ResNet-v2. All mentioned studies were able to utilize many times more extensive databases for model training. Furthermore, because of a higher soft tissue contrast [32,33], MRI is able to represent the anatomy in more detail. Also, the characteristics of the study population differed fundamentally.

The absolute prediction error showed no significant difference between younger and older subjects. Female subjects were significantly estimated to be younger, which can be attributed to the higher mean age of the female subjects in this study, where the age of older patients was underestimated by the model.

The estimated uncertainty was higher for subjects younger than 53 years. This could have been caused by lower variability in the body regions important for age estimation, because a differentiation between more similar representations is also more challenging. A comparable phenomenon was shown in previous work regarding age estimation based on brain MRI [15].

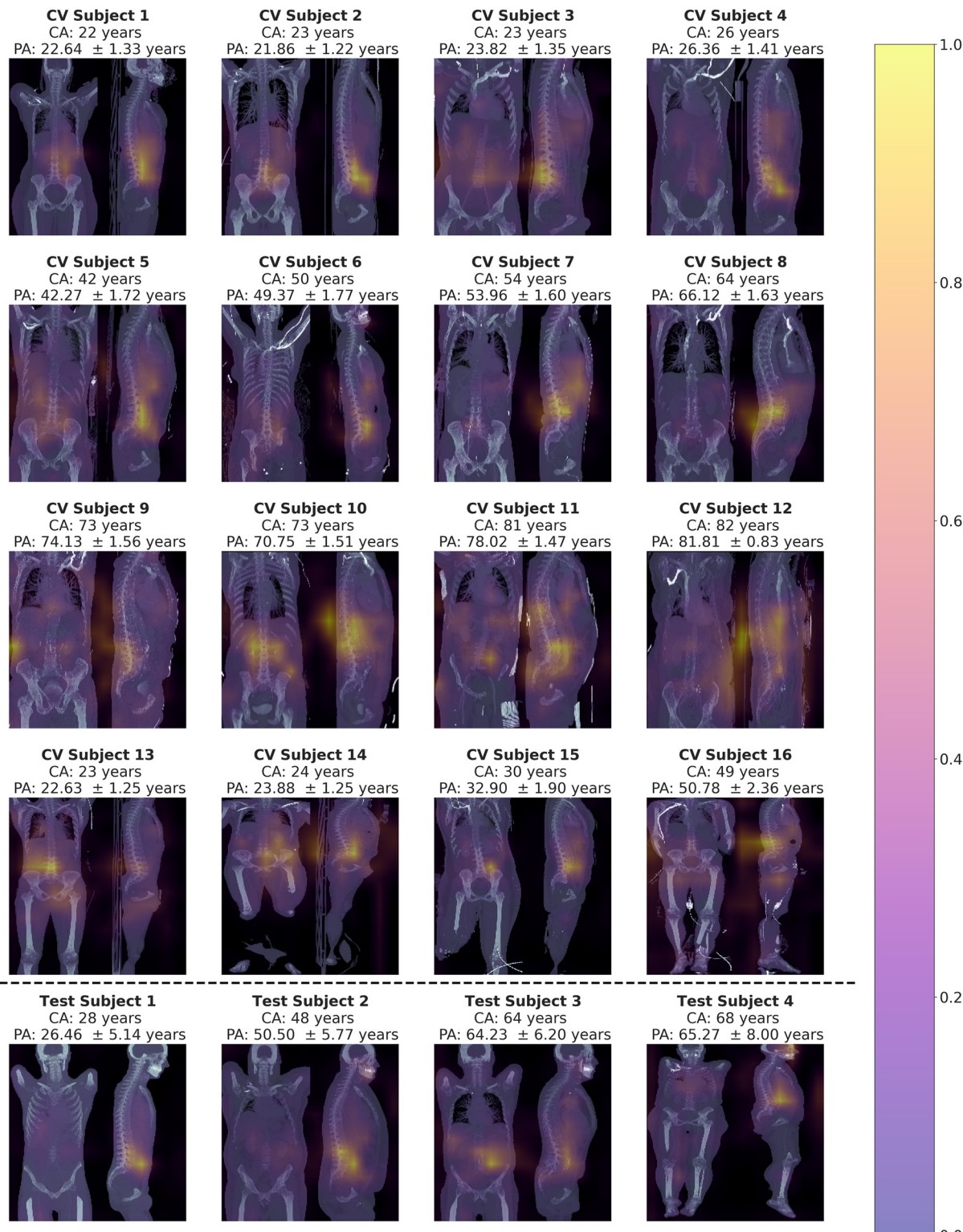

**Fig 6. Exemplary Score-CAM saliency maps of selected subjects, with their chronological age (CA) and predicted age (PA) ± uncertainty given.** First row: Young subjects. Second row: Middle aged subjects. Third row: Older subjects. Fourth row: Subjects, whose images differ from the majority of other subjects. For example, in contrast the legs are included to various degrees. Over all subjects, the pattern of the sagittal MIP was more consistent than the frontal MIP. Fifth row: Subjects from the test dataset. The lumbar spine and abdominal aorta were highlighted as regions important for age estimation. The pattern was consistent over different age groups and even expressed in unusual subjects that differed from the majority. Higher values on the colorbar indicate higher importance.

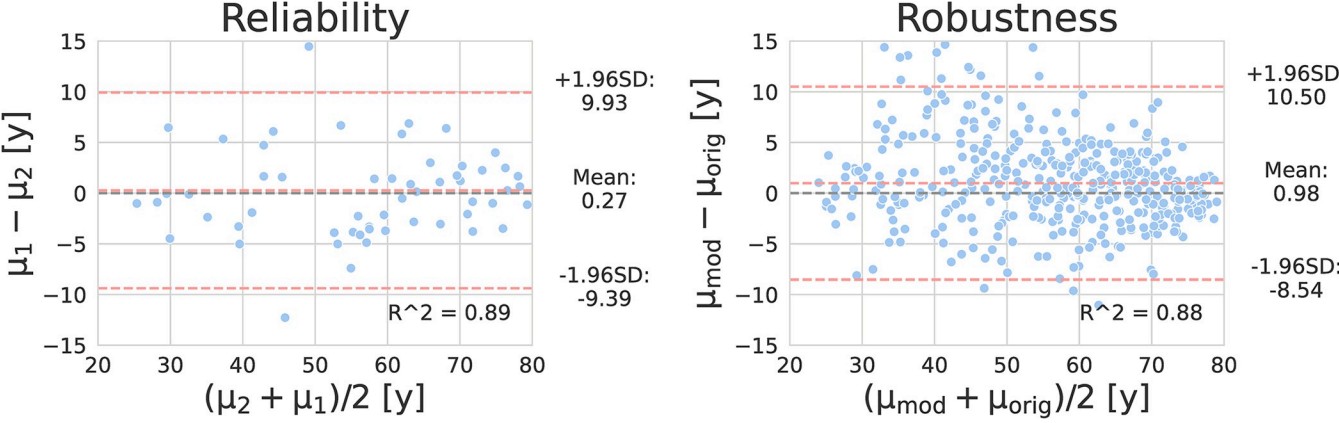

**Fig 7. Reliability and Robustness of the trained model.** Left: Bland-Altman plot of model predictions for different studies of the same subject. Right: Bland-Altman plot of model predictions for original and modified images. A significant correlation between the age predictions for different studies of the same subject ($R^2 = 0.89$) was observed. A significant correlation between the age predictions for the original and modified images ($R^2 = 0.88$) was observed.

The significant negative correlation between subject number and estimated uncertainty raises the question, if the estimated uncertainty is purely aleatoric or represents a non-negligible epistemic component from a lack of training data.

In this work, emergency trauma CT-scans images generated in the context of clinical routine in the emergency room of the University Hospital Tübingen were used for model training. That lead to, in comparison to other studies, a smaller and more heterogenous dataset. Prediction accuracy could be improved in a more controlled cohort and imaging setup.

A slightly higher MAE of 6.50 years was obtained on the publicly available test dataset. Also, the estimated uncertainty was significantly higher and had a positive correlation with the MAE.

Again, the characteristics of the study population and their image data in the test dataset differed from the training and validation dataset.

One cause is the difference of indications for performing imaging between the datasets. While in the training and validation dataset, subjects received a CT-examination to rule out polytrauma, subjects in the test dataset were examined for diagnosis, staging or recurrence of different tumor entities.

This, on the one hand, influenced the study population. The share of female subjects was higher in the test dataset. In addition, the subjects in the test dataset were, on average, older than their counterparts in the training and validation dataset. This could have caused the network to underestimate the age of, especially the female, subjects, resulting in a negative mean prediction error.

On the other hand, it also influenced the used field of view in the image data, with the head and lower body being included more frequently in the test dataset. Our model sometimes seemed to struggle with images where the legs were included. This could be attributed to the fact, that important regions in the abdomen were smaller due to the standardized image size.

However, our proposed model showed overall good ability to generalize to independent, unseen data. Moreover, significant and very strong correlations between the model predictions for different studies of the same patient and between predictions for original and perturbed images from the test dataset were observed. This suggests, that the model predictions are reliable, and robust against small, random variations in the input data.

When uncertainty is considered, high uncertainty can indicate problems with the model (epistemic uncertainty, e.g. from a lack of training data) or that the underlying data is

challenging to interpret (aleatoric uncertainty, e.g. from ambiguous or noisy input data). In our study, the significantly higher uncertainty in age estimations of younger subjects can be interpreted as, at least partly, aleatoric uncertainty due to the chosen formulation in the negative log-likelihood loss. Also, the estimated uncertainty was significantly higher on the test dataset and for the perturbed images, which may indicate, that our model is able to detect unusual variations in the input data and express its uncertainty about its predictions on said inputs. Considering the significant relationship with the MAE, the estimated uncertainty also indicated the possibility of higher prediction errors, as it was intended to. Nevertheless, further research is needed to understand the exact causes of uncertainty in our proposed model.

The finding, that sufficiently accurate reliable and robust age estimation, that also generalizes to unseen data, is possible using a heterogenous dataset acquired in the clinical routine, is important. The integration of machine learning models into safety-critical infrastructure like the healthcare system requires said models to perform reliably and be robust. This poses a key challenge for deploying machine learning in a real world scenario [34].

A major assumption for machine learning is that training and test data are drawn from the same feature space and have the same distribution [35–37]. This is, however, not the case in most real-world scenarios [35].

For example, it was shown by Zech, Badgeley [38], that the specific hospital system where a radiograph was acquired could be reliably identified using a Convolutional Neural Network. It was assumed, that subtle site-specific characteristics in the presented image data made the site identification possible, leading to poor generalization in other tasks [38]. Therefore, characteristics of image data acquired in previous studies might differ from that of image data generated in a specific clinical setting. Such distribution shifts can impede the performance of models [34].

However, while some data characteristics might change, other underlying features remain stable, which is also indicated by the performance of our trained model on the publicly available dataset. Also, the generated saliency maps seemed similar to those of the training and validation dataset, which may imply, that similar image regions were chosen for the age prediction.

Furthermore, it would be desirable to adapt to new data, while also retaining previous knowledge. Transfer learning, aiming at transferring knowledge from a source task onto the process of solving a related task [39], could pose a solution. For example, a hospital willing to use artificial intelligence could be able to fine-tune a pretrained model on their characteristic data acquired in the clinical routine.

Saliency maps visualizing regions that are important for age prediction can help to explain the predictions and build trust in the model [40,41]. In this study, generated saliency maps for the model on the validation and the test dataset seemed plausible. The most important regions for age estimation appeared to be the lumbar spine and abdominal aorta, which are both subject to extensive changes in the aging process [42–45].

While saliency maps can help to explain model predictions, it has to be noted that the research into explainability of deep learning models is still evolving [46] and the interpretation of saliency maps remains challenging, with explanations being not necessarily reliable or even misleading [47].

In this work, we used image data of subjects, who received a non-pathological trauma scan in the emergency room and were thus assumed to approach subjects drawn from a healthy cohort. There was no other clinical information available, such as past medical history. It is unclear, how this impacts age estimation. A clinical correlation should be investigated in further research.

Furthermore, the image data of chosen subjects was assessed retrospectively. A relatively small sample size was used to train a deep learning model. The results of our work should be evaluated prospectively in the future on a more diverse cohort.

We foresee future clinical applications mainly in the area of age-dependent risk scores. Further research is needed to generate a clear clinical impact.

## 5. Conclusion

It was shown in this study, that accurate deep learning-based automated age estimation using CT-images of thorax and abdomen acquired in the clinical routine is feasible. The model also generalized well to previously unseen image data and is proven to be robust and reliable.

The results of this study are valuable for further research regarding automated age estimation and highlight the future possibilities of transfer learning applications in medical image analysis.

This study demonstrates furthermore how methods of uncertainty estimation and saliency analysis can be used in a deep learning-based framework to improve the interpretability of and gain valuable knowledge from its predictions.

## Acknowledgments

We thank the IZKF graduate school of the University of Tübingen for their support.

## Author Contributions

**Conceptualization:** Bjarne Kerber, Sergios Gatidis.

**Data curation:** Bjarne Kerber.

**Formal analysis:** Bjarne Kerber.

**Methodology:** Bjarne Kerber, Tobias Hepp, Thomas Küstner, Sergios Gatidis.

**Software:** Bjarne Kerber.

**Visualization:** Bjarne Kerber, Sergios Gatidis.

**Writing – original draft:** Bjarne Kerber.

**Writing – review & editing:** Bjarne Kerber, Tobias Hepp, Thomas Küstner, Sergios Gatidis.

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
