## [Decision Letter · Decision Letter 0]

23 May 2023

PONE-D-23-10836Deep learning-based age estimation from clinical Computed Tomography image data of the thorax and abdomenPLOS ONE

Dear Dr. Kerber,

Thank you for submitting your manuscript to PLOS ONE. After careful consideration, we feel that it has merit but does not fully meet PLOS ONE’s publication criteria as it currently stands. Therefore, we invite you to submit a revised version of the manuscript that addresses the points raised during the review process. The reviewers have identified several issues that need to be addressed in detail, and the manuscript changed accordingly. Also, please find the comments of the academic/handling editor below, that suggest a more in depth validation of the method.

We look forward to receiving your revised manuscript.

Kind regards,

Peter Homolka

Academic Editor

PLOS ONE

Journal Requirements:

3. Please remove your figures from within your manuscript file, leaving only the individual TIFF/EPS image files, uploaded separately. These will be automatically included in the reviewers’ PDF.

Additional Editor Comments:

- As also stated by Reviewer #1, the limitations should more clearly be concentrated in a subsection at the end of the discussion.

In the revision also pls consider the following issues (1st minor, 2nd major):

- Age estimation is most demanding but likely most useful for minors under and about the age of 20. If these are excluded, this is a strict limitation of the study that needs to be reflected in the title, because many potential readers will be interested in the age determination of younger persons, e.g. for forensic issues. So a more appropriate title would be, e,g, "Deep learning-based age estimation from clinical Computed Tomography image data of the thorax and abdomen in the adult population"

- Regarding the usually performed and necessary validation steps refer to Recht, M. P. et al. Integrating artificial intelligence into the clinical practice of radiology: challenges and recommendations. Eur Radiol 30, 3576-3584, doi:10.1007/s00330-020-06672-5 (2020), Technical validation section, and make sure to verify robustness, reproducibility, and generalizability appropriately. This does not seem possible with the limited data set used for verification. This needs to be done appropriately, especially robustness and generalizability. This needs to checked and discussed.

Reviewers' comments:

Reviewer's Responses to Questions

**Comments to the Author**

1. Is the manuscript technically sound, and do the data support the conclusions?

Reviewer #1: Yes

Reviewer #2: Yes

2. Has the statistical analysis been performed appropriately and rigorously? 

Reviewer #1: Yes

Reviewer #2: Yes

3. Have the authors made all data underlying the findings in their manuscript fully available?

Reviewer #1: No

Reviewer #2: No

4. Is the manuscript presented in an intelligible fashion and written in standard English?

Reviewer #1: Yes

Reviewer #2: Yes

5. Review Comments to the Author

Reviewer #1: The authors present a deep neural network to predict the chronological age from thoracic and abdominal CT scans in an adult cohort; the model obtains an MAE of around 5-6 years on the independent external test set.

The paper has overall sound quality. Its strong point is that it used a larger clinical dataset and an external dataset for testing. As a result, it increases the trust in the study.

There are no major weak points, in my opinion. However, the paper could benefit mainly from rewriting the Discussion to be more concise. Currently, many small paragraphs do not fit well together, and the limitations are somehow 'hidden' instead of being stated clearly at the end of the section, as is usual, at least in medical papers.

Abstract

- "Recently, it has been shown that accurate chronological age prediction is possible using deep learning-based approaches trained on healthy cohorts." Please be more specific about the modality of this approach, It sounds like your study has already been conducted.

Introduction

- l42-44: This sentence does not fit and could be removed/merged with the next paragraph.

- l57: there is a gap since deep learning as such was not mentioned/introduced.

- l57-66: these paragraphs feel somehow disconnected.

- l86: please give references for MR+age estimation.

Methods

- l106-109: I am unsure if "rule out significant trauma" is correct. Please explain. Also, does this mean that you assumed that emergency room patients are (otherwise) healthy? If so, please be more specific about it, already in the Abstract.

- l109-110: Why were patients <20 or >85 years excluded? I would have expected <18 years or maybe <21 years-- and no exclusion of older adults.

- l111: Please give a patient flowchart including ALL train and test cohorts exclusion criteria. E.g., were multiple images of the same patient excluded?

- l130: 3mm seems large, was this part of the inclusion criteria? What was the inplane resolution?

- l146: This is 'critical' since this preprocessing converts the 3D data into 2D so that it could lose information. Since it is now virtually a set of localizers, it would be reasonable to compare your model to existing 2D models in the Introduction section. Please also add an image with, say, three examples to get a visual impression of the projections. It is part of the visualization, but a dedicated figure upfront might help.

- l149: Why offline augmentations were performed instead of the very common online augmentations?

- l151: I need clarification about the network. I thought the 2D shape of 224x224 was chosen to fit the default ImageNet resolution, yet, you state that you cropped 56x112 pixels and resized them between 1 and 1.5. This resizing would result in images of different sizes between 56x112 and around 85x172? So the network was fed with different sizes? Or was this resized to 224x224? If so, it seems to be a very small crop size?

- l154-166: Personally, I'm not too fond of splitting the description of network architecture and implementation details. Please merge the relevant sections (2.4,2.5,2.7) into a single section, explaining all parameters needed for reproducing the network. Add hardware and software details at the end of this paragraph or in an extra section before statistical analysis. Then continue with 2.6, which can be regarded as post-processing. About the loss: Is the given loss function the only loss function used for training?

- l185: There is no mention that the splitting was performed along patients instead of images?

- l192: I misread this at first; please state clearly that you do not retrain but re-use the trained models for training, i.e., you create a small ensemble.

Results

- l201: I am missing relative errors. While it is customary to only report MAEs in a machine learning context, in the medical context, a relative error is more reasonable since, say, a difference of 10 years might be different for an 18 year-old than for a 85 year-old. Please compute the relative errors and also plots them in a suitable way.

- l214: Subanalyses, especially one wrt kernels and scanners, were not mentioned before. Could you add more details to the Methods part?

- l222: There is something wrong with the stated prediction error. 6.24 maybe?

- l230-235: These paragraphs feel very disconnected. Please rewrite. Maybe it can help to put all results into a table and compute them for CV and test?

- l241: "than in the ... validation dataset?". This isn't very clear (also in the plots) since you actually never split off a validation set but used only CV.

- Figure 3: so the network never predicted age <20 years and >81 years? This seems strange; can you explain?

Discussion:

- l283: I do not think your MAE is competitive with the UK biobank study. The R2 is, but what does a linearly measured correlation tell us about prediction quality?

- l290: Why is the population different? The UK Biobank should have 'random' patients-- isn't this the same as in the emergency room?

- l295-299: I think you mean "body regions", however, I find this paragraph hard to follow. Besides, could your uncertainty be a statistical effect of cutting off patients <20 years (the same for >85 years)?

- l305: Can you cite other studies with a more homogenous population? Both studies from above should also be very heterogeneous.

- l324: This could maybe have been avoided by normalized wrt pixel/voxel size.

- Code/Data

Can you share the code and, if possible, also the model? Since the test data is freely available, this would help with reproducibility.

Reviewer #2: The authors present a deep learning approach to estimate age from a chest CT image. The model is trained on a dataset of 1653 images from patients being evaluated for significant trauma at a single center and tested on a publicly available Ct dataset of 421 subjects. Interpretability techniques and uncertainty estimation are used to provide context about features of aging and to add robustness to the model. Overall the model accurately predicts age and the study is properly done. The authors should be commended for the high prediction accuracy and the ability to ingest difficult 3-D data in the deep learning approach. I just have a few methodological concerns and requests for additional data.

Major comments

1) Interpretation of grad-cam heatmaps has been called into question in recent studies. Would dial back claims that score-cam can improve interpretability/identify specific features important for age estimation. Also, it’s unclear whether interpretability analyses were done systematically. Did you quantify how often certain regions of the body were highlighted by these heatmaps? If not, would make it clear that the interpretability results are a qualitative assessment by the study authors and not a rigorous investigation.

2) The authors should make their code and model weights publicly available to facilitate future research. It would also be helpful for the authors to make their predicted ages for the training/testing datasets publicly available and any other derived data as far as possible.

3) The publicly available testing dataset is not adequately described. Who are these individuals? Why were they getting CT? Did they have any characteristic information available? A Table 1 describing the two cohorts would be very helpful to better interpret the results of the study.

4) Please clarify the limitations of the study in the Discussion. Would include things like

a. Small sample size for training to develop a deep learning model

b. Unclear demographic characteristics of these cohorts – will need to evaluate on a more diverse cohort

c. Unclear clinical applicability, how specifically will clinicians/researchers use this model?

d. Single center, retrospective study – needs to be evaluated prospectively

e. Unclear comorbidity history and how this impacts age estimates

Abstract

The authors state that aging “… leading to morphological change that can be assessed on medical imaging sch as computed tomography scans” – have others shown this already? If so, please highlight such studies in the discussion and potentially compare your model to these existing models. If not, then this is the hypothesis of your work and not a statement of the background.

Methods

Why wasn’t data augmentation performed online? Why were 49 augmented images chosen? Did this number impact generalization performance/training time?

Small typo – ”ResNet18-architecture pretrained on the by Pytorch”

Results

The first line of the results is a little unclear to me – “achieved MAE after 5-fold cross validation” – is this the average MAE across the 5 validation folds? Or is this the result on the independent testing data? Would be helpful to define mean prediction error and mean absolute error.

Typo in the comparison of prediction error between female and male subjects (-3.57 +/- 8.09 vs. 62.46)

In the uncertainty estimation paragraph, I think the result comparing uncertainty vs. prediction error is the most important and should be better highlighted.

Discussion

Wouldn’t compare these results with study focusing on whole-body MRI data since there is a substantially different N and different modality here.

Have any others attempted to estimate age from CT? This will be good to highlight instead

Do you think the higher estimated uncertainty in the test set was due to generalization error or due to differences in patient characteristics? What were the differences in patient characteristics between the two populations?

Any breakdown available of reasons CTs were performed in the testing dataset?

Again, would dial back claims about “features” being used for age prediction

You state in the discussion that uncertainty “indicated the possibility of higher prediction errors as it was intended to” – These results were not clearly stated – did uncertainty correlate with error?

6. PLOS authors have the option to publish the peer review history of their article (what does this mean?). If published, this will include your full peer review and any attached files.

Reviewer #1: No

Reviewer #2: No

---

## [Author Response · Author response to Decision Letter 0]

12 Jul 2023

Dear Prof. Homolka, dear Reviewers,

we appreciate the opportunity to submit our revised manuscript (PONE-D-23-10836) titled

“Deep learning-based age estimation from clinical Computed Tomography image data of the thorax and abdomen in the adult population”

Bjarne Kerber, Tobias Hepp, Thomas Küstner, Sergios Gatidis.

We thank you for taking the time to edit and review our work and are grateful for the insightful comments. We addressed the concerns raised in the decision letter and incorporated the suggestions into our revised manuscript. Please find our response below.

Yours sincerely

The authors

Response

Editor Comments and Responses

• Comment 1: As also stated by Reviewer #1, the limitations should more clearly be concentrated in a subsection at the end of the discussion.

Response: We thank the Reviewers and the Editor for pointing out, that the limitations should be written more concisely. We improved our manuscript accordingly.

• Comment 2: Age estimation is most demanding but likely most useful for minors under and about the age of 20. If these are excluded, this is a strict limitation of the study that needs to be reflected in the title, because many potential readers will be interested in the age determination of younger persons, e.g. for forensic issues. So a more appropriate title would be, e,g, "Deep learning-based age estimation from clinical Computed Tomography image data of the thorax and abdomen in the adult population“

Response: The editor is right, that this distinction should be made to avoid confusion. Unfortunately, we had no access to a sufficient number of pediatric cases. The revised manuscript was given a more appropriate title.

• Comment 3: Regarding the usually performed and necessary validation steps refer to Recht, M. P. et al. Integrating artificial intelligence into the clinical practice of radiology: challenges and recommendations. Eur Radiol 30, 3576-3584, doi:10.1007/s00330-020-06672-5 (2020), Technical validation section, and make sure to verify robustness, reproducibility, and generalizability appropriately. This does not seem possible with the limited data set used for verification. This needs to be done appropriately, especially robustness and generalizability. This needs to be checked and discussed.

Response: We are grateful that the editor has addressed this important point. The literature cited by the editor provides valuable recommendations for the technical validation of machine learning models to be used in healthcare with respect to robustness, reproducibility, and generalizability. We addressed these challenges by conducting additional experiments and incorporating the results obtained in our revised manuscript. In particular, we performed two analyses regarding variation in age estimation. First, we processed the images from the test group using perturbation methods such as noise injection and random resizing and compared the age predictions to the predictions for the original images. Second, for individuals in the test set with multiple scans, we tested the reproducibility of the age prediction of our model for different scans. We found that our trained model, in addition to generalizing well to unseen data, has high overall robustness and reproducibility. The relatively large size of our test data set allowed us to conduct these additional experiments.

Reviewer #1

Abstract

• Comment 1: "Recently, it has been shown that accurate chronological age prediction is possible using deep learning-based approaches trained on healthy cohorts." Please be more specific about the modality of this approach, It sounds like your study has already been conducted.

Response: Thank you for raising this point. The reviewer is right, that this statement needs clarification. We have revised the manuscript accordingly. 

Introduction

• Comment 2: l42-44: This sentence does not fit and could be removed/merged with the next paragraph.

Response: Agree. We have revised the manuscript accordingly. 

• Comment 3: l57: there is a gap since deep learning as such was not mentioned/introduced.

Response: Thank you. We have revised the manuscript accordingly.

• Comment 4: l57-66: these paragraphs feel somehow disconnected.

Response: Agree. We have revised the manuscript accordingly.

• Comment 5: l86: please give references for MR+age estimation.

Response: Thank you for pointing that out. We included the requested references.

Methods

• Comment 6: l106-109: I am unsure if "rule out significant trauma" is correct. Please explain. Also, does this mean that you assumed that emergency room patients are (otherwise) healthy? If so, please be more specific about it, already in the Abstract.

Response: Thank you for your valuable suggestion. The reviewer is right, that this needs clarification. We will improve our manuscript in this regard. We included emergency room patients, who received a trauma CT where no pathological findings were reported, into our work.

• Comment 7: l109-110: Why were patients <20 or >85 years excluded? I would have expected <18 years or maybe <21 years-- and no exclusion of older adults.

Response: Thank you for raising this important point. In the initial dataset, there were only small numbers of patients younger than 20 years and older than 85 years, so that we restricted the age range to the subjects, where we had a sufficient number of examples.

• Comment 8: l111: Please give a patient flowchart including ALL train and test cohorts exclusion criteria. E.g., were multiple images of the same patient excluded?

Response: Thank you for this valuable comment. We will clarify on the issue. Only a single study for each patient was included. The requested flowchart was included in our revised manuscript. 

• Comment 9: l130: 3mm seems large, was this part of the inclusion criteria? What was the inplane resolution?

Response: Thank you for this question. Examinations were drawn from the time period between 2010 and 2019, were this was a standard slice thickness for emergency trauma scans in our institution. 

• Comment 10: l146: This is 'critical' since this preprocessing converts the 3D data into 2D so that it could lose information. Since it is now virtually a set of localizers, it would be reasonable to compare your model to existing 2D models in the Introduction section. Please also add an image with, say, three examples to get a visual impression of the projections. It is part of the visualization, but a dedicated figure upfront might help.

Response: Thank you for raising an important point here. We chose this approach to make model training and predictions more computationally feasible, while still preserving some of the 3D-information. We referenced 2D-approaches in the introduction (Langner and Wikström, Raghu and Weiss) and compared them in the discussion. We added the requested input examples to our revised manuscript. We agree that using a 3D-model could potentially improve CT-based age estimation. Overcoming associated challenges regarding algorithm design and computational feasibility will be part of future research.

• Comment 11: l149: Why offline augmentations were performed instead of the very common online augmentations?

Response: Thank you for this question. We used offline data augmentation to save training time and avoid CPU-bottlenecks. However, the wording is misleading. It was meant, that data augmentation was performed for the training process, not during it. We clarified the statement in the revised manuscript.

• Comment 12: l151: I need clarification about the network. I thought the 2D shape of 224x224 was chosen to fit the default ImageNet resolution, yet, you state that you cropped 56x112 pixels and resized them between 1 and 1.5. This resizing would result in images of different sizes between 56x112 and around 85x172? So the network was fed with different sizes? Or was this resized to 224x224? If so, it seems to be a very small crop size?

Response: Thank you for raising this point. Every input image had the standard ImageNet resolution of 224x224 pixel. The cropped patches were randomly resized, shifted and rotated, while the rest of the image was filled with zeros. An example is shown in Figure 2.

• Comment 13: l154-166: Personally, I'm not too fond of splitting the description of network architecture and implementation details. Please merge the relevant sections (2.4,2.5,2.7) into a single section, explaining all parameters needed for reproducing the network. Add hardware and software details at the end of this paragraph or in an extra section before statistical analysis. Then continue with 2.6, which can be regarded as post-processing. About the loss: Is the given loss function the only loss function used for training?

Response: Agree. We revised our manuscript accordingly. The loss function mentioned is the only loss function used for training.

• Comment 14: l185: There is no mention that the splitting was performed along patients instead of images?

Response: Thank you for this question. The splitting was performed along patients. This is now mentioned in the revised manuscript.

• Comment 15: l192: I misread this at first; please state clearly that you do not retrain but re-use the trained models for training, i.e., you create a small ensemble.

Response: Thank you for raising this point. We trained a new model for each fold of the cross-validation. The predictions on the validation datasets were single predictions from the trained model of said validation fold. The predictions on the previously unseen test dataset were average predictions of the five trained models. We will explicitly state that in our revised manuscript.

Results

• Comment 16: l201: I am missing relative errors. While it is customary to only report MAEs in a machine learning context, in the medical context, a relative error is more reasonable since, say, a difference of 10 years might be different for an 18 year-old than for a 85 year-old. Please compute the relative errors and also plots them in a suitable way.

Response: Thank you for your suggestion. In response, we added a graph showing that there is no monotonic relation between chronological age and MAE. Therefore relative errors would only be of limited value.

• Comment 17: l214: Subanalyses, especially one wrt kernels and scanners, were not mentioned before. Could you add more details to the Methods part?

Response: Agree. We revised our manuscript accordingly.

• Comment 18: l222: There is something wrong with the stated prediction error. 6.24 maybe?

Response: Agree. We revised our manuscript accordingly.

• Comment 19: l230-235: These paragraphs feel very disconnected. Please rewrite. Maybe it can help to put all results into a table and compute them for CV and test?

Response: Thank you for this suggestion. We revised the mentioned paragraph.

• Comment 20: l241: "than in the ... validation dataset?". This isn't very clear (also in the plots) since you actually never split off a validation set but used only CV.

Response: Thank you for raising this point. We clarified this sentence in our revised manuscript.

• Comment 21: Figure 3: so the network never predicted age <20 years and >81 years? This seems strange; can you explain?

Response: Thank you for this valuable question. The network was trained using the gaussian negative log likelihood loss, were the network predicts the mean and standard deviation of a gaussian distribution, in which the true age should lie. This could be the explanation for this behavior, because a network has a tendency to avoid extreme values to obtain a smaller loss. 

Discussion:

• Comment 22: l283: I do not think your MAE is competitive with the UK biobank study. The R2 is, but what does a linearly measured correlation tell us about prediction quality?

Response: Thank you for this very relevant question. We agree and revised our manuscript accordingly. The core difference between the two works is the much larger age span with a higher standard deviation of our study population in comparison to the UK Biobank study. The significant R2 measured the correlation between the predicted age and the true age, which is supposed to be a linear relationship. The R2 was chosen to show, that the predictions of our model are strongly and significantly correlated to the true age, which shows the quality of the predictions of our model. We incorporated a Bland-Altman-plot of the network predictions and true age to underpin the prediction quality visually.

• Comment 23: l290: Why is the population different? The UK Biobank should have 'random' patients-- isn't this the same as in the emergency room?

Response: Thank you for raising this interesting point. The study population was chosen from the emergency room, because a large number of patients received a non-pathological trauma scan. The population of the emergency room differs from the general population, because e.g. more young people are involved in car accidents, while older people are more likely to suffer trauma from falls, which also leads to a certain selection of individuals, why the UK Biobank study is a true population study.

• Comment 24: l295-299: I think you mean "body regions", however, I find this paragraph hard to follow. Besides, could your uncertainty be a statistical effect of cutting off patients <20 years (the same for >85 years)?

Response: Thank you for this suggestion. The reviewer is right, that the paragraph needs to be worded better. 

We don’t believe that cutting off patients would influence the uncertainty. There was a medium strong correlation between the number of original training examples and uncertainty, which might indicate, that the model uncertainty reduces, when more training images have been seen. The uncertainty on the “edges” of our age distribution is lower than in the middle, because there are plenty of training examples, while there are less examples around the mean of our age distribution.

• Comment 25: l305: Can you cite other studies with a more homogenous population? Both studies from above should also be very heterogeneous.

Response: Thank you for raising this point. The UK Biobank was in this respect more homogenous, that the same, standardized imaging protocol was used and subjects were preselected, while we used a more clinically realistic dataset of actual patients.

• Comment 26: l324: This could maybe have been avoided by normalized wrt pixel/voxel size.

Response: Thank you for this suggestion. We suspect that the inclusion of the legs changed the image data and made important regions smaller, but we chose to include said images anyways to make the model more robust.

Code/Data

• Comment 27: Can you share the code and, if possible, also the model? Since the test data is freely available, this would help with reproducibility.

Response: Agree. We will make the requested data available. The code for preprocessing, the model, its weights and the Score-CAM implementation can be found at https://github.com/BjarneKerber/age_estimation. 

Reviewer #2 

Major comments

• Comment 1: Interpretation of grad-cam heatmaps has been called into question in recent studies. Would dial back claims that score-cam can improve interpretability/identify specific features important for age estimation. Also, it’s unclear whether interpretability analyses were done systematically. Did you quantify how often certain regions of the body were highlighted by these heatmaps? If not, would make it clear that the interpretability results are a qualitative assessment by the study authors and not a rigorous investigation.

Response: Thank you for this important comment. We made clear, that the interpretability results are just qualitative to avoid confusion. We revised our manuscript accordingly.

• Comment 2: The authors should make their code and model weights publicly available to facilitate future research. It would also be helpful for the authors to make their predicted ages for the training/testing datasets publicly available and any other derived data as far as possible.

Response: Thank you for your comment. We will make the requested data available. The code for preprocessing, the model, its weights and the Score-CAM implementation can be found at https://github.com/BjarneKerber/age_estimation.

• Comment 3: The publicly available testing dataset is not adequately described. Who are these individuals? Why were they getting CT? Did they have any characteristic information available? A Table 1 describing the two cohorts would be very helpful to better interpret the results of the study.

Response: Thank you for your suggestion. We added additional information about the dataset in our manuscript. The whole dataset is described in detail “A whole-body FDG-PET/CT Dataset with manually annotated Tumor Lesions. Sci Data. 2022 Oct 4;9(1):601. doi: 10.1038/s41597-022-01718-3. PMID: 36195599; PMCID: PMC9532417”.

• Comment 4: Please clarify the limitations of the study in the Discussion. Would include things like

a. Small sample size for training to develop a deep learning model

b. Unclear demographic characteristics of these cohorts – will need to evaluate on a more diverse cohort

c. Unclear clinical applicability, how specifically will clinicians/researchers use this model?

d. Single center, retrospective study – needs to be evaluated prospectively

e. Unclear comorbidity history and how this impacts age estimates

Response: Thank you for your valuable suggestions. We incorporated your requested points into the limitations section of our revised manuscript.

Abstract

• Comment 5: The authors state that aging “… leading to morphological change that can be assessed on medical imaging such as computed tomography scans” – have others shown this already? If so, please highlight such studies in the discussion and potentially compare your model to these existing models. If not, then this is the hypothesis of your work and not a statement of the background.

Response: Thank you for pointing this out. It is a scientific consensus, that CT image data is able to reflect the morphological change occurring during the aging process. In our work we want to show, that automated age estimation is possible using said fact. To our knowledge, no models exist for automated age estimation on thorax/abdomen CT image data generated in a clinical context. We clarified this statement in the revised manuscript.

Methods

• Comment 6: Why wasn’t data augmentation performed online? Why were 49 augmented images chosen? Did this number impact generalization performance/training time?

Response: Thank you for this question. Offline data augmentation saved training time and helped to avoid CPU-bottlenecks. The number of augmented images was chosen empirically, because even the small network used needed strong regularization, which we wanted to achieve by using extensive data augmentation.

• Comment 7: Small typo – ”ResNet18-architecture pretrained on the by Pytorch”

Response: Thank you for pointing this out. We fixed the typo.

Results

• Comment 8: The first line of the results is a little unclear to me – “achieved MAE after 5-fold cross validation” – is this the average MAE across the 5 validation folds? Or is this the result on the independent testing data? Would be helpful to define mean prediction error and mean absolute error.

Response: Thank you for your suggestion. You are right, the computed MAE is the average across the 5 validation folds. We improved the wording in the referenced section.

• Comment 9: Typo in the comparison of prediction error between female and male subjects (-3.57 +/- 8.09 vs. 62.46)

Response: Thank you for pointing this out. We fixed the typo.

• Comment 10: In the uncertainty estimation paragraph, I think the result comparing uncertainty vs. prediction error is the most important and should be better highlighted.

Response: Thank you for this proposal. We revised our manuscript accordingly.

Discussion

• Comment 11: Wouldn’t compare these results with study focusing on whole-body MRI data since there is a substantially different N and different modality here.

Response: The reviewer is right, that the mentioned study was able to use a much more extensive database and that the modality influences prediction quality, because e.g. MRI has a better soft tissue contrast. We chose to compare our work with this study, because the approach is similar and discussed the differences you correctly mentioned. We furthermore compared the results with the age estimation on the CheXpert dataset, because there was also a similar approach used.

• Comment 12: Have any others attempted to estimate age from CT? This will be good to highlight instead

Response: Thank you for your suggestion. To our knowledge, this is the first study to use thorax/abdomen CT image data for automated age estimation in the adult population. We thus compare our results with other studies using MRI or radiography data.

• Comment 13: Do you think the higher estimated uncertainty in the test set was due to generalization error or due to differences in patient characteristics? What were the differences in patient characteristics between the two populations?

Response: Thank you for asking this very relevant question. There was a higher MAE on the test dataset. The estimated uncertainty had a significant positive correlation with the MAE on both the training and validation and on the test dataset. We intended the estimated uncertainty to act as an indicator of possible higher prediction error. The uncertainty is probably influenced by differing characteristics between the training and the test dataset, that also influenced the generalization error. The differences in patient characteristics are summarized in the discussion and include a different mean age, a higher percentage of female subjects as well as a different indication for performing a CT-scan, namely the suspicion of malignancy.

• Comment 14: Any breakdown available of reasons CTs were performed in the testing dataset?

Response: Thank you for this question. CTs in the test dataset are derived from PET/CT examinations, which were performed because of suspected malignant tumors. The whole dataset is described in detail in “A whole-body FDG-PET/CT Dataset with manually annotated Tumor Lesions. Sci Data. 2022 Oct 4;9(1):601. doi: 10.1038/s41597-022-01718-3. PMID: 36195599; PMCID: PMC9532417”.

• Comment 15: Again, would dial back claims about “features” being used for age prediction

Response: Agree, we revised our manuscript accordingly. 

• Comment 16: You state in the discussion that uncertainty “indicated the possibility of higher prediction errors as it was intended to” – These results were not clearly stated – did uncertainty correlate with error?

Response: Thank you for this interesting question. There was a weak, but significant correlation between uncertainty and absolute prediction error (l.269). The relationship between prediction error and uncertainty is also shown in Fig. 4.

---

## [Decision Letter · Decision Letter 1]

15 Aug 2023

PONE-D-23-10836R1Deep learning-based age estimation from clinical Computed Tomography image data of the thorax and abdomen in the adult populationPLOS ONE

Dear Dr. Kerber,

Thank you for submitting your manuscript to PLOS ONE. After careful consideration, we feel that it has merit but does not fully meet PLOS ONE’s publication criteria as it currently stands. Therefore, we invite you to submit a revised version of the manuscript that addresses the points raised during the review process. Please consider the valid comments by reviewer #2, that require your attention. However, since it is one review to attend to, I ask for a minor revision only and look forward to receiving it shortly.

We look forward to receiving your revised manuscript.

Kind regards,

Peter Homolka

Academic Editor

PLOS ONE

Journal Requirements:

Reviewers' comments:

Reviewer's Responses to Questions

**Comments to the Author**

1. If the authors have adequately addressed your comments raised in a previous round of review and you feel that this manuscript is now acceptable for publication, you may indicate that here to bypass the “Comments to the Author” section, enter your conflict of interest statement in the “Confidential to Editor” section, and submit your "Accept" recommendation.

Reviewer #1: All comments have been addressed

Reviewer #2: (No Response)

2. Is the manuscript technically sound, and do the data support the conclusions?

Reviewer #1: Yes

Reviewer #2: Yes

3. Has the statistical analysis been performed appropriately and rigorously? 

Reviewer #1: Yes

Reviewer #2: Yes

4. Have the authors made all data underlying the findings in their manuscript fully available?

Reviewer #1: Yes

Reviewer #2: Yes

5. Is the manuscript presented in an intelligible fashion and written in standard English?

Reviewer #1: Yes

Reviewer #2: Yes

6. Review Comments to the Author

Reviewer #1: Thank you for your revision, the corrections improved the manuscript considerably. I suppose the github link will be active after acceptance. I have no further comments.

Reviewer #2: The authors have provided a revised version of their manuscript to assess chronologic age from chest and abdominal CT. The manuscript is much improved, but I still have additional questions.

1) Thanks to the authors for adding a limitations section. I believe there are additional limitations missing.

- Small sample size for training a deep learning model

- Need to evaluate the model on a more diverse cohort – single center and demographics were not available

2) There has been a recent study investigating the performance of deep learning to estimate age from chest CT – would include this as a reference in your Discussion:

Azarfar G, Ko SB, Adams SJ, Babyn PS. Deep learning-based age estimation from chest CT scans. Int J Comput Assist Radiol Surg. 2023 Jul 7. doi: 10.1007/s11548-023-02989-w. Epub ahead of print. PMID: 37418109

3) Abstract, Line 26 – would remove “hallmarks of aging,” it is unclear whether score-cam identified image regions are truly hallmarks of aging without a more detailed quantitative analysis.

4) Abstract, Line 34-35 – would also remove “The trained model proved to be robust and reliable” – without external validation, it’s hard to claim this. Also, it’s unclear whether uncertainty estimation improved interpretability of the model. This was not clearly demonstrated with your data.

5) Line 107 – were all 2,419 images from unique patients? How many patients were originally represented?

6) Line 110 – 354 images is a substantial proportion of the cohort. Did you need to exclude those >85? This could make age estimation for individuals in their 80s subject to greater uncertainty. Would add additional justification for this. Removing pediatric cases is easier to justify since the images themselves will be substantially different.

7) How was the single study for each subject chosen? (Line 114)

8) Figure 5 – Would consider testing how uncertainty changes the variability in predictions instead of the absolute prediction error. For example, high uncertainty should really imply that predictions are variable, not necessarily that they are skewed in a particular direction away from the conditional mean.

9) Line 129 – “Subjects with a scan positive for malignancy were excluded” – could be really interesting to see if these individuals had higher predicted age than their chronologic age? This would suggest that the model is able to identify malignancies.

10) Section 3.3 – still need to make clear that these are qualitative observations. It’s unclear how many images were observed, and whether these patterns are likely to apply to the full testing dataset.

7. PLOS authors have the option to publish the peer review history of their article (what does this mean?). If published, this will include your full peer review and any attached files.

Reviewer #1: No

Reviewer #2: No

---

## [Author Response · Author response to Decision Letter 1]

1 Sep 2023

Dear Prof. Homolka, dear Reviewers,

we appreciate the opportunity to submit our revised manuscript (PONE-D-23-10836R2) titled

“Deep learning-based age estimation from clinical Computed Tomography image data of the thorax and abdomen in the adult population”

Bjarne Kerber, Tobias Hepp, Thomas Küstner, Sergios Gatidis.

Thank you again for your effort in editing and reviewing our work. We greatly appreciate the valuable comments you have provided. We have addressed the concerns expressed in the decision letter and revised our manuscript accordingly. Please find our response below.

Yours sincerely

The authors

Response

Reviewer #2 

• Comment 1: Thanks to the authors for adding a limitations section. I believe there are additional limitations missing.

- Small sample size for training a deep learning model

- Need to evaluate the model on a more diverse cohort – single center and demographics were not available

Response: Thank you for this important suggestion. The desired changes were incorporated into the revised manuscript.

• Comment 2: There has been a recent study investigating the performance of deep learning to estimate age from chest CT – would include this as a reference in your Discussion:

Azarfar G, Ko SB, Adams SJ, Babyn PS. Deep learning-based age estimation from chest CT scans. Int J Comput Assist Radiol Surg. 2023 Jul 7. doi: 10.1007/s11548-023-02989-w. Epub ahead of print. PMID: 37418109

Response: Thank you for sharing this interesting publication. We discussed the findings.

• Comment 3: Abstract, Line 26 – would remove “hallmarks of aging,” it is unclear whether score-cam identified image regions are truly hallmarks of aging without a more detailed quantitative analysis.

Response: Thank you for this important comment. We revised the manuscript accordingly.

• Comment 4: Abstract, Line 34-35 – would also remove “The trained model proved to be robust and reliable” – without external validation, it’s hard to claim this. Also, it’s unclear whether uncertainty estimation improved interpretability of the model. This was not clearly demonstrated with your data.

Response: Thank you for raising this point. Technical validation was performed according to the suggestions of Recht et al. on an external dataset as stated in section 2.6. Thus, we feel justified to state the robustness and reliability of our model. Furthermore, we found a significant positive correlation between model uncertainty and prediction error. Hence, a higher uncertainty could signal a higher prediction error, thus improving the interpretability of our model predictions.

(Recht, M. P., Dewey, M., Dreyer, K., Langlotz, C., Niessen, W., Prainsack, B., & Smith, J. J. (2020). Integrating artificial intelligence into the clinical practice of radiology: challenges and recommendations. European radiology, 30, 3576-3584).

• Comment 5: Line 107 – were all 2,419 images from unique patients? How many patients were originally represented?

Response: Thank you for this question. Not all images were from unique patients. Duplicate subjects were excluded as stated in the exclusion criteria. There were more than 2000 original subjects.

• Comment 6: Line 110 – 354 images is a substantial proportion of the cohort. Did you need to exclude those >85? This could make age estimation for individuals in their 80s subject to greater uncertainty. Would add additional justification for this. Removing pediatric cases is easier to justify since the images themselves will be substantially different.

Response: Thank you for making this important point. The 354 images mentioned are distributed across the whole age range from 0 to 100 years and male and female subjects. In the initial dataset, there were only small numbers of patients younger than 20 years and older than 85 years compared to subjects in the chosen age range. In some age groups, there were less than 5 subjects, sometimes only a single male or female subject. We thus focused on the consecutive age range where we had a sufficient number of examples for both sexes. 

• Comment 7: How was the single study for each subject chosen? (Line 114)

Response: Thank you for this question. The first study of each subject was chosen.

• Comment 8: Figure 5 – Would consider testing how uncertainty changes the variability in predictions instead of the absolute prediction error. For example, high uncertainty should really imply that predictions are variable, not necessarily that they are skewed in a particular direction away from the conditional mean.

Response: Thank you for this interesting suggestion. The uncertainty predicted by our model approaches the aleatoric uncertainty, which is subject to the inherent noise and ambiguity in the data, while a epistemic uncertainty component e.g. from a lack of training data cannot be ruled out. You are right, that in highly noisy or ambiguous inputs, the uncertainty should measure higher variability of outputs. Because we are only able to collect a single datapoint per patient by predicting its age, we unfortunately cannot measure the variability of predictions. In higher uncertainty, the error is expected to be higher, which we use as a surrogate marker to evaluate the predicted uncertainty. 

• Comment 9: Line 129 – “Subjects with a scan positive for malignancy were excluded” – could be really interesting to see if these individuals had higher predicted age than their chronologic age? This would suggest that the model is able to identify malignancies.

Response: Thank you for this valuable suggestion. The analysis of performance on subjects with a known disease is an interesting field we plan to explore in future work.

• Comment 10: Section 3.3 – still need to make clear that these are qualitative observations. It’s unclear how many images were observed, and whether these patterns are likely to apply to the full testing dataset.

Response: Thank you for this important comment. We revised our manuscript accordingly.

---

## [Decision Letter · Decision Letter 2]

4 Oct 2023

Deep learning-based age estimation from clinical Computed Tomography image data of the thorax and abdomen in the adult population

PONE-D-23-10836R2

Dear Dr. Kerber,

We’re pleased to inform you that your manuscript has been judged scientifically suitable for publication and will be formally accepted for publication once it meets all outstanding technical requirements.

Kind regards,

Peter Homolka

Academic Editor

PLOS ONE

Additional Editor Comments (optional):

All comments have been addressed appropriately. I would like to thank the authors for their contributions

Best wishes

Reviewers' comments:

Reviewer's Responses to Questions

**Comments to the Author**

1. If the authors have adequately addressed your comments raised in a previous round of review and you feel that this manuscript is now acceptable for publication, you may indicate that here to bypass the “Comments to the Author” section, enter your conflict of interest statement in the “Confidential to Editor” section, and submit your "Accept" recommendation.

Reviewer #1: All comments have been addressed

2. Is the manuscript technically sound, and do the data support the conclusions?

Reviewer #1: Yes

3. Has the statistical analysis been performed appropriately and rigorously? 

Reviewer #1: Yes

4. Have the authors made all data underlying the findings in their manuscript fully available?

Reviewer #1: Yes

5. Is the manuscript presented in an intelligible fashion and written in standard English?

Reviewer #1: Yes

6. Review Comments to the Author

Reviewer #1: Thank you for your revision.

7. PLOS authors have the option to publish the peer review history of their article (what does this mean?). If published, this will include your full peer review and any attached files.

Reviewer #1: No

---

## [Editor Report · Acceptance letter]

30 Oct 2023

PONE-D-23-10836R2 

Deep learning-based age estimation from clinical Computed Tomography image data of the thorax and abdomen in the adult population 

Dear Dr. Kerber:

I'm pleased to inform you that your manuscript has been deemed suitable for publication in PLOS ONE. Congratulations! Your manuscript is now with our production department. 

Kind regards, 

on behalf of

Dr. Peter Homolka 

Academic Editor

PLOS ONE